A behavioral and genetic study of multiple paternity in a polygamous marine invertebrate, Octopus oliveri

Ylitalo Heather 1
Oliver Thomas A. 2
Fernandez-Silva Iria 1 3
Wood James B. 1
Toonen Robert J. toonen@hawaii.edu rjtoonen@gmail.com 1
1 Hawai‘i Institute of Marine Biology, University of Hawai‘i at Mānoa , Kāne‘ohe , HI , United States of America
2 Department of Oceanography, University of Hawai‘i at Mānoa , Honolulu , HI , United States of America
3 Department of Genetics, Biochemistry and Immunology, University of Vigo , Vigo , Spain
Archie Elizabeth
Electronic publication date: 2019 Jun 7
Publication date: 2019
Volume: 7
Electronic Location ID: e6927
Received 2018 Oct 30; Accepted 2019 Apr 7
Copyright: ©2019 Ylitalo et al.
Copyright year: 2019
Copyright holder: Ylitalo et al.
License: This is an open access article distributed under the terms of the Creative Commons Attribution License, which permits unrestricted use, distribution, reproduction and adaptation in any medium and for any purpose provided that it is properly attributed. For attribution, the original author(s), title, publication source (PeerJ) and either DOI or URL of the article must be cited.
License URL: https://creativecommons.org/licenses/by/4.0/

Keywords: Microsatellite, Mating behavior, Cephalopod, Reproduction, Polygyny

Funding: NOAA National Marine Sanctuaries Program award 2005-008/66882 European Union’s Seventh Framework Programme FP7/2007-2013 REA 302957 600391 FELLOWSEA: Campus do Mar International Fellowship Program University of Hawai‘i, Department of Biology Edmondson Grant National Science Foundation NSF-OA#1416889 This work was supported primarily by a NOAA National Marine Sanctuaries Program award (MOA grant No. 2005-008/66882) to Robert J. Toonen and two Marie Curie Actions of the European Union’s Seventh Framework Programme (FP7/2007-2013) under REA grant agreements 302957 and 600391 (FELLOWSEA: Campus do Mar International Fellowship Program) to Iria Fernandez-Silva. Additional funding was provided to Heather Ylitalo through the University of Hawai‘i, Department of Biology Edmondson Grant, and to Robert J. Toonen through the National Science Foundation (NSF-OA#1416889). The funders had no role in study design, data collection and analysis, decision to publish, or preparation of the manuscript.

==============================
Octopus oliveri is a widespread and common rocky intertidal cephalopod that mates readily in the laboratory, but for which mating behavior has not been reported previously. Four sets of behavioral experiments were recorded wherein three males, small, medium & large in varying order, were introduced to each of six females, for a total of 24 individual females and 12 individual males utilized in the experiments. Video analysis shows that successful mating occurred in each of the mount, reach and beak-to-beak positions. Mating was observed for all males, regardless of size relative to the female, or order of introduction. Females showed preference for the first male to which they were introduced in experimental pairings rather than any specific male trait, and mating time increased significantly with increasing female size. Five novel microsatellite markers were developed and used to test paternity in the eleven broods resulting from these experimental pairings. We found skewed paternity in each brood, with early male precedence and male size being the best predictors of parentage. Multiple paternity was observed in every experimental cross but was estimated to be comparatively low in the field, suggesting that sperm limitation might be common in this species. We saw no evidence of direct sperm competition in Octopus oliveri, but larger males produced significantly more offspring. This study contributes to the growing research on cephalopod mating systems and indicates that octopus mating dynamics might be more variable and complex than thought previously.

Introduction

Multiple paternity, or the presence of numerous males fertilizing offspring in one brood, is common across many taxa, in both vertebrates and invertebrates (Toonen, 2004; Daly-Engel, Grubbs & Holland, 2006; Cutuli et al., 2013). In mating systems where multiple paternity occurs, it is often common to have high rates of sperm competition, which occurs when sperm from two or more males compete to fertilize the ova of a female (Birkhead & Møller, 1998; Birkhead & Pizzari, 2002).

Within the Cephalopoda, sperm competition has been observed in a variety of squid and cuttlefish species in the form of mate guarding, sneaker males, sperm flushing and increased sperm allocation (Wada et al., 2010; Squires et al., 2015; Naud et al., 2016). In octopuses, sperm competition is believed to occur due to the presence of multiple mating, two oviducts in which to store sperm, and long-term sperm storage capabilities (Birkhead & Møller, 1998; Hanlon & Messenger, 1998; Wigby & Chapman, 2004; Morse et al., 2018). Yet, mate guarding and sneaker behavior has been described in few species (Cigliano, 1995; Tsuchiya & Uzu, 1997; Huffard, Caldwell & Boneka, 2010).

Male sperm precedence is the nonrandom utilization of one male’s sperm over another’s (Birkhead & Møller, 1998). This can occur through female cryptic choice within the oviduct of the female, overt female rejection of sperm packets, or through male displacement of previously placed sperm packets by rival males. In nature, some animals show first male sperm precedence (Tennessen & Zamudio, 2003), where the first males to inseminate a female produce the most fertilized gametes, while others exhibit a “last in, first out” strategy (Birkhead & Møller, 1998). Among octopus, evidence of sperm precedence has only been reported in the southern blue-ringed octopus, Hapalochlaena maculosa (Morse et al., 2015; Morse et al., 2018) and an unnamed species of pygmy octopus in which the mechanism of sperm competition remains unknown (Cigliano, 1995). For both species, males spent more time mating with a female that had previously mated. Cigliano (1995) concluded that this pattern suggested that the second male was somehow removing or displacing sperm from a previous male. Based on these studies, there might be a trend among octopuses of the last male siring more offspring than the first male to mate with the female.

Three previous studies have been conducted using microsatellites to determine whether multiple paternity was present in octopus broods, one with Graneledone boreopacifica (Voight & Feldheim, 2009), one with Octopus vulgaris (Quinteiro et al., 2011), and the last with Octopus minor (Bo et al., 2016). Each of these studies confirmed that multiple paternity was occurring in these species, however they did not observe mating prior to collecting the eggs, so it is unknown if mating behavior affected fertilization success. We wanted to test whether successive males also showed any evidence of sperm competition with previous mates in O. oliveri. We also asked whether any conspicuous male trait, such as body size or aggression, predicted the observed mating success. For example, large body size can be a predictor in determining mating success not only in octopuses, but across many taxa (Andersson & Iwasa, 1996; Birkhead & Møller, 1998; Huffard, Caldwell & Boneka, 2010). In addition, larger body size might be an indicator to females of genetic superiority in survivability and trigger female choice, so we also wanted to test whether size is a predictor of mating success.

The male octopus has a modified third right arm called the hectocotylus, which he uses to transfer sperm packets (spermatophores) to the female (Anderson, Mather & Wood, 2013). A sperm mass is encapsulated along with an ejaculatory organ in each spermatophore. The tip of the hectocotylus is characterized by a ligula and calamus. The male passes a spermatophore down the groove of the hectocotylized arm to either of the two distal oviducts of the female. As the spermatophore is passed down through the penis and into the groove of the hectocotylus, osmotic pressure begins to force water through the outer tunic of the spermatophore. The male reaches into the mantle of the female with his hectocotylus and transfers the spermatophore to the distal oviduct where the ejaculatory process begins (Anderson, Mather & Wood, 2013). The sperm mass is released from the spermatophore and it travels up the oviduct and is stored in the spermathecae in either of the two oviducal glands, along with the sperm from previously mated males (Wells, 1978; Mann, 1984; Hanlon & Messenger, 1998; Wodinsky, 2008). Females can mate with multiple males before laying eggs and can store sperm for up to 10 months in some species (Mangold, 1987). The eggs become fertilized as they travel through the oviducal gland and down the oviduct (Forsythe & Hanlon, 1988). As with many other species, Octopus oliveri females lay several strands of eggs, each with multiple eggs, which they protect for approximately one month before hatching (Ylitalo, Watling & Toonen, 2014).

This study describes the mating behavior of a minimally studied intertidal cephalopod, Octopus oliveri, and tests the following questions: are all mating attempts successful? If not, do females differentially reject copulation attempts based on male size or mate order? Is multiple paternity present in this species? If so, what are the ratios of paternity for each male, and can we detect evidence of sperm precedence in this species or differential success of males among fertilized egg strands?

Materials and Methods

Animal collection and husbandry

Octopus oliveri individuals were collected from Kaka‘ako Waterfront Park (21°17′33.6″N 157°51′51.5″W), and Kewalo Basin Marina (21°29′09.57″N 157°85′77.35″W Honolulu, Hawai‘i, USA in the fall of 2010 through the summer of 2013 (over 100 individuals collected, 70 different trips into the field). These collections were permitted under Hawai‘i State Department of Land and Natural Resources, Division of Aquatic Resources Special Activities Permits #HIMB-SAP2010, HIMB-SAP2011, HIMB-SAP2012 & HIMB-SAP2013. Two to three people would walk along the rock wall during the evening hours for one to three hours (between 7 pm–12 am) with a flashlight. When an octopus was found, it was collected by hand and transferred to a five-gallon bucket. The males and females were kept in separate buckets. Adult octopuses were weighed on a platform scale (wet weight) and transferred to tanks on Coconut Island, Kāne‘ohe. Each octopus was housed in an individual tank (38 cm × 21 cm × 23.5 cm) with a piece of coral or PVC pipe for shelter and a plastic well-ventilated lid. These tanks were then placed in a large outdoor tank at the Hawai‘i Institute of Marine Biology (HIMB) with constant saltwater flow (30 gallons per minute) and ambient ocean temperature (mean temperature 25.5 °C ± 0.6). The octopuses were fed frozen shrimp or live crabs daily until satiated and the tanks were cleaned after each feeding. Water temperature records were obtained though NOAA Tides and Currents databases from the station located closest to the collection sites in Honolulu (Station ID 1612340) and at Coconut Island (Station ID 1612481). Females collected that laid eggs before experimental trials began were considered to be representative of natural populations (controls) and were not used in mating experiments. They were allowed to brood their eggs until natural senescence and their eggs were tested as non-experimental indicators of paternity in the wild. We do not report an animal use permit because cephalopods are not regulated by IACUC at the University of Hawai‘i at the time of this study.

Mating experiments

For each series of mating experiments (four in total) six females and three males were chosen randomly from the available pool of collected octopuses (raw data provided in Data S1). Each female was paired with each of the three males (one male at a time) for a total of 18 individual mating trials per set of experiments. The males were chosen with maximum variation in size, one being the largest, one smallest, and one midsize. Each of the six females had a different order of mates (i.e., female 1 with male A, B, C, female 2 with male B, C, A etc.) allowing for every possible pairing combination (Fig. 1).

Figure 1 Design of octopus breeding experiment.

The top row represents the females used in the experiment while the bottom row is representative of the males of varying sizes (small medium, and large). Each pair is connected with the mating order listed at the top near the female. For example, female i mated with male A 1st, male B 2nd, and male C 3rd. Six separate females and three separate males were used for each of the four mating experiments that were run. Due to the death of some individuals, only 62 mating events between the octopuses were recorded (raw data presented in Data S1). A tissue biopsy from the arm of each female and male following the mating experiments provided the DNA for paternity testing.

All mating trials occurred at night, as this species is nocturnal (Ylitalo, Watling & Toonen, 2014). Three 15-gallon (61 cm × 32 cm ×  32 cm) tanks were set up with constant seawater flow (2 gallons per minute) and separated by black plastic to ensure that adjacent pairs did not influence the other octopuses. Sessions were recorded using a 6 LED USB PC Web Camera with the infrared filter removed. A camera was mounted 100 centimeters above each tank (measured from the floor of the tank). A 48-LED illuminator infrared light was placed in front of each tank to illuminate the video without disturbing the octopuses.

The female was always placed in the tank first and allowed to settle for approximately 10 min. Then, the male was introduced and the trial would begin. Three pairs were filmed simultaneously, each pair with its own camera, during each experimental night. Trials lasted at least two hours and would end when the mating pair separated. Also, if a female tried to escape from the tank three times, the trial was ended as it was predicted the female would have escaped the male in the field. In some cases, this would mean the trial would last less than two hours. Videos were analyzed after all trials were completed.

Spermatogenesis after mating has been explored in several cephalopods, often with sperm production occurring immediately following copulation (Van Heukelem, 1976; Hanlon & Ament, 1999). However, the rates of sperm production vary across individuals. Given this knowledge, the males were allowed to rest one day (a conservative estimate based on the literature) between sessions to allow for sperm regeneration.

The trial history of each female was recorded to analyze whether mate order or mate size influenced the observed mating success. Mating success was described as the amount of time a male spent mating with a female and the number of times he was able to complete the arch and pump movements.

If an animal died before a trial could be carried out, the pairing could not occur and was therefore removed from the total number of possible recorded trials. Sixty-two trials were completed and over 125 h of video were analyzed twice by the same observer (H Ylitalo) and once by another observer (J Yamada) to ensure continuity between evaluations of behavior. Three central behaviors were recorded: mating, fighting (agonistic behavior) and resting. A trial was considered successful when any or all of these behaviors between the two octopuses were recorded. Within these general categories, more specific interactions were described as follows.

Mating was described as the period starting with the male approaching the female and feeling around her mantle and arms, attempting to insert the hectocotylus. When the hectocotylus was inserted, the male would begin arch and pump movements. During the “arch” movement, the male lifted the groove on the hectocotylized arm to the mantle, lining it up with the penis inside the mantle cavity, giving the male a hunched appearance. This was followed by the “pump”, when the male inflated the mantle in a deep respiratory movement and exhaled explosively, sending the spermatophore down the ridge of the arm and into the oviducal gland of the female (Wells & Wells, 1972; Wodinsky, 2008). The number of times a male completed each arch and pump movement was recorded as well as the time between first inserting the hectocotylus to the first arch and pump.

Fighting (agonistic behavior) was described as the period when at least one octopus appeared to be trying to escape the other. Writhing arms (grappling), suckers pulling on skin (arm pulling), and biting were observed, however no inking was ever noted. During fighting, the hectocotylus was clearly not inserted in the female, but physical contact was necessary for fighting to be recorded. In some instances, fighting would result in mating (generally in the mount position), while in others the octopuses would separate and try to escape or a resting period would begin. Fighting was not always followed by mating, nor was mating followed by fighting, indicating forced copulation was not occurring.

Resting behavior was described as the period of time when neither octopus was touching the other, but could be moving around the tank, or lying still so long as they were not interacting. Individuals had to be apart from each other and the male could not have any arm inside the female for resting to be recorded.

In addition to these three main behaviors, any instances of female behavior that could be perceived as female choice were recorded. For instance, if a female was seen to approach the male to begin mating, if a female did not mate with one male but did mate with others, or if a female was seen to overtly remove a sperm packet during any trial, the act was recorded.

Behavioral data analyses

Differences between the amount of time spent mating, fighting or resting between the first, second or third male to mate with the female were tested using the non-parametric Friedman rank test (FRX). The Friedman rank test was used because it is based on rank and median, rather than mean and variance, and is appropriate for repeated measures, so it is robust to the non-independence of males among replicates. Only females that mated with each of the three males in their set were included in this analysis (n = 15).

To analyze the effect of mate size on mating, fighting, and resting duration, the non-parametric Kruskal–Wallis [KW(x)] test was used (R code for all analyses provided in Supplemental Information 2). For this analysis, all trials (n = 46) were included except those of the females that did not mate in any of their three trials (n = 16). Male size relative to female was calculated by dividing female weight by male weight (grams) (see Supplemental Information 2). Males that were within 15% of female weight were considered equal in size, those more than 15% below female weight were classified as small, and those at least 15% above were classified as large males. Similarly, male size relative to average male size (n = 12 males) in the sampled population and female size relative to average female size (n = 24 females) in the sampled population were calculated.

Finally, female mating choice among males was tested using the Chi-square test (χ2) on the subsample of trials (n = 27) in which females successfully mated with all three paired males and were observed exhibiting behavior resembling female choice. All statistical tests were run in R (Version 0.99.902 R Studio, Inc.) (R Core Team, 2013).

Genetic analyses of paternity

Arm tip muscle tissue was collected from 11 adult females (those that laid eggs; 8 experimental and 3 wild caught) and 9 adult males (the males that were mated with the brooding females). Egg strings from each clutch were collected one or two days before hatching and fixed in separate vials of 90% ethanol. Thirty-four individual eggs were sampled from 9–12 randomly selected strands from each of the 11 broods of females. The number of eggs sampled was determined using power analysis (see Supplementary Information). Eggs were randomly sampled from the top, middle, and bottom section of the egg strand and their locations were recorded. The paralarvae were almost fully developed at this time to provide the most DNA possible. DNA extractions were performed using the HotSHOT protocol on each embryo and adult muscular tissue sample (Truett et al., 2000).

Microsatellite loci developed for Octopus vulgaris and Graneledone boreopacifica (Greatorex et al., 2000; Voight & Feldheim, 2009; Quinteiro et al., 2011) were tested for use in Octopus oliveri, however they all failed to amplify a product. Therefore, species-specific microsatellite primers were designed for Octopus oliveri (Fernandez-Silva et al., 2013). Initially, 48 putative loci (raw sequences provided in Supplemental Information 3) were tested, but after screening, only the 5 best sets of primers (Table 1) were optimized (Selkoe & Toonen, 2006). The three-tailed primer method described by Gaither et al. (2009) was then used in PCR amplification.

Table 1 Summary of microsatellite markers used for this study.

Novel species-specific microsatellite markers developed for Octopus oliveri and used in this study, the primer and tagged sequences, annealing temperature, size and levels of polymorphism. Sequences and additional loci not used in this study are provided in the Supplementary Information.

Locus	Motif	Primer Sequence (5′-3′)	Ta	Size range (bp)	NA	HO	HE	Freq of nulls	
Octoli_003	(TAGA)12	F: T1-GCACGTTGTACGCGATTC	62	154–200	11	0.888	0.856	0.018	
		R: ATATGCATGAAGACGCAACTC				
Octoli_007	(TATG)12	F: T2-CGCAGACGAGGAATCAATAG	62	152–184	9	0.718	0.816	0.063	
		R: GGAGAACAGACACAAGAACACAG				
Octoli_017	(TATG)8	F: T2-AGCAACACGATGGCCTCTAC	60	180–202	5	0.569	0.521	0.048	
		R: AGTCCAACAAGCTTCGATCC					
Octoli_022	(TGA)21	F: T1-AGCCATGTGGTTGAGAACG	60	239–287	14	0.943	0.902	0.022	
		R: GCGTGCCTCTCTTCATCAG					
Octoli_023	(GAT)20	F: T3-GCCATGAATTCCAAGTAACTAACC	60	160–199	15	0.856	0.846	0.007	
		R: CATCGTCATACGCCATCATC					
Notes.

T1 PET-5′-GGCTAGGAAAGGTTAGTGGC-3′

T2 6-Fam-5′-TCATACATGTCTCTCAGCGTAAAC-3′

T3 VIC-5′-GACTATGGGC GTGAGTGCAT-3′

T4 NED-5′-ACCAACCTAGGAAACACAG-3′

Ta Annealing temperature (°C)

NA Number of alleles

HO Observed heterozygosity

HE Expected heterozygosity

Two primer mixes were prepared for each individual to be genotyped. Primer mix A consisted of 10 µl each of 100 mM primer Octoli_3R, Octoli_7R, Octoli_10R, Octoli_11R, fluorescent yellow (NED), red (PET), green (VIC), and blue (6-fam) dye. In addition, there were 2.5 µl of 100 mM primer Octoli_3F-T1, Octoli_7F-T2, Octoli_10F-T4, and Octoli_11F-T3 (Table 1). The rest of the mixture comprised of 410 µl of RNAse-free water (H2O). Primer mix B used the same ratio of solutions as listed above for Primer mix A, however primers Octoli_17, Octoli_18, Octoli_22, and Octoli_23 were used. Octoli_10, Octoli_11, and Octoli_18 were not used in the final analysis due to multiple non-Mendelian peaks in amplification, but were kept in the primer mixes to ensure no differences in amplification among samples would occur. Each individual PCR reaction mix contained 3 µl 2×Multiplex MasterMix (from a QIAGEN Multiplex PCR kit), 0.6 µl 10×Primer mix as outlined above, 1.4 µl RNAse free water, and 1 µl template DNA (1:10 dilution of extraction) for a final reaction volume of 6 µl.

PCR amplification was completed on a Bio-Rad iCyler as follows: 95 °C for 15 min (1 cycle), followed by 35 cycles of 95 °C for 30 s, 60 °C or 62 °C (see Table 1) for 90 s, 72 °C for 60 s, followed by a final extension of 72 °C for 30 min. Amplified PCR products were visualized on an Applied Biosystems 3730X Genetic Analyzer at the University of Hawai’i at Manoa, and genotyped using Geneious version 6.7.1 (Biomatters, (Kearse et al., 2012) following guidelines from Selkoe & Toonen (2006).

Parentage and multiple paternity analyses using genetic data

The maternal genotypes from each of the 11 broods were compared with the embryo genotypes manually to ensure that at least one maternal allele was found for each embryo at each locus, confirming Mendelian inheritance (after Selkoe & Toonen, 2006). Then, after excluding the maternal alleles one can make a conservative estimate of the number of sires contributing to a brood by using the single-locus minimum (SLM) method of counting the number of non-maternal alleles at each locus in the progeny, dividing the largest number by two (assuming all males are heterozygotes), and rounding up (Toonen, 2004; Jones, 2005).

The program GERUD v. 2.0 was then used to evaluate the frequency of multiple paternity within broods based on population allele frequencies to find the most likely umber of paternal genotypes (Jones, 2005; Croshaw, Peters & Glenn, 2009). GERUD was also used to calculate the expected exclusion probability for each locus and for the combined loci (see Supplemental Information 4). Two of the experimental broods required the locus Octoli_17 to be excluded from the analyses, because inclusion consistently caused the GERUD software to crash.

Parentage was assessed using the maximum likelihood ratio program in CERVUS v. 3.0 (Marshall et al., 1998; Slate, Marshall & Pemberton, 2000; Jones et al., 2010). The likelihood ratio is the probability that the candidate parent is the true parent compared with the probability of an alternate unrelated candidate parent. The program uses this ratio to determine the most likely father given a known maternal genotype, a set of candidate paternal genotypes, and the brood genotypes. CERVUS incorporates genotyping error, unsampled candidate parents, and missing genotypes into the program analysis. Both strict (95%) and relaxed (80%) confidence in paternal assignment was used, but did not alter the interpretation of the data, so only the 95% assignment was used here (as recommended by (Marshall et al., 1998).

Any offspring not assigned paternity at 95% confidence were then rerun through GERUD to find potential paternal genotypes from the wild, under the assumption that wild males who mated with females before collection sired the unassigned offspring. GERUD also calculates how many offspring are assigned to each wild type male. To corroborate the number of eggs assigned to paternal genotypes generated by GERUD, CERVUS was run again using only unassigned eggs (at a 95% confidence level). Finally, the program fmm was used to assess the frequency of multiple mating in the natural population of Octopus oliveri using the genotypes of broods of non-experimental females (Neff, 2002). This program considers the number of loci, the number of alleles and their frequencies, and reproductive skew. These results were used to corroborate multiple paternity through the SLM and GERUD methods and to extrapolate rates of multiple paternity in wild populations.

Differences in the ratio of offspring sired by experimental males were tested using chi-squared test (χ2), whereas ANOVA and Pearson’s product-moment correlations were used to test for differences in: mating time, male order, male size, number of arch and pumps, and frequency of removed sperm packets, on the number of eggs sired by each experimental male. The best model of predictors was calculated using marginal likelihood ratio tests and AIC (Akaike Information Criterion) model selection tables (see Supplemental Information 4).

Results

Mating behavior in Octopus oliveri

Of 62 behavioral observations during attempted crosses between 36 individuals (24 females and 12 males), 46 trials included mating. This number was reduced from the expected total (24 experimental females introduced to each of 3 males = 72 attempted crosses) because a few females died, escaped, or laid eggs before completing all three experimental mating trials.

As with most octopuses, the mating behavior observed both between individuals and among multiple mating bouts within individuals was varied (Wells & Wells, 1972; Huffard, Caldwell & Beneka, 2008). However, some general patterns emerge. The average time it took for the male to approach the female and begin mating was 18 min (standard error [σ] = 17 min), with the shortest amount of time being 8 s and the longest 1 h and 7 min. No obvious courtship was seen in either behavior or body patterns for either male or female octopuses in these trials. The male would touch the female all over her mantle and arms while searching for the oviduct with his hectocotylus for approximately 30 s to one minute. Most mating occurred in either the arm reach or mount position (sensu (Wells, 1978), however in 12 trials, beak-to-beak mating (Rodaniche, 1984) was observed (Fig. 2). The most a male was able to arch and pump in one mating trial was 74 times, the least was 5 times, with an average of 25 times during a single mating session (σ = 18 arch and pumps). The average time between each arch and pump was 2 min and 12 s (σ = 1 min 26 s). Despite the lack of obvious courtship leading up to copulation, during mating itself, the male was generally a dark brown-red color and the female was a pale white, although this was not always seen.

Figure 2 Images of mating behaviors captured from video.

Video stills of four mating pairs of Octopus oliveri in the beak-to-beak mating position. Females (♀) and males (♂) are indicated in each mating pair pictured A–D.

Mating would end when either partner would detach from the other (generally the female), either to begin fighting or resting. The longest uninterrupted mating duration was 1 h and 33 min, but in general, each trial was characterized by many short bouts of repeated mating, the shortest being approximately 1 min in duration. The average time spent mating (all short bouts added together) per trial was 1 h (σ = 45 min). In the 16 trials where no mating occurred, variable times and combinations of both fighting and resting were observed. The data from these final 16 trials were used in the size analysis but not in mating precedence analyses, where only females who mated in all three trials were used.

Male precedence effect

We saw no evidence of a male precedence effect in our behavioral observations of mating. Fifteen of the 24 experimental females successfully mated with all three experimental males in these trials. From the perspective of the female (see Supplemental Information 4), there were no significant differences in the rate or duration of mating, fighting, or resting as successive males were presented (mating FRX = 0.43, p = 0.82, fighting FRX = 1, p = 0.61, resting FR X = 1, p = 0.81, n = 15). Nor was there a difference in the number of arch and pumps seen during successive mating trials (FRX = 0.32, p = 0.81, n = 15). Likewise, there was no significant difference in the time it took for males to begin the first arch and pump between successive mating trials (FRX = 1.56, p = 0.46, n = 9). The same is true of the individual behavioral patterns of the males (see Supplemental Information 4), in which no significant difference was found in response to successive females to which each was introduced (mating: FRX = 0.4, p = 0.81, fighting: FRX = 0.93, p = 0.61, resting: FRX = 0.4, p = 0.61, number of arch and pumps: FRX = 0.43, p = 0.85, time from start of mating to first arch and pump: FRX = 2, p = 0.37, n = 9).

Effect of size on mating behavior

There was no significant trend (see Supplemental Information 4) between relative male size and the likelihood of mating (KW(x) = 0.31, p = 0.85, n = 52), resting (KW(x) = 1.22, p = 0.54, n = 52) or fighting (KW(x) = 0.06, p = 0.97, n = 52) with a given female. Likewise, male size (either absolute or relative to the female) did not appear to affect the number of times a male would arch and pump (KW(x) = 3.21, p = 0.2, n = 52). Male size relative to other males (see Supplemental Information 4), also had no significant effect on mating (KW(x) = 1.92, p = 0.38), fighting (KW(x) = 2.32, p = 0.31), resting (KW(x) = 0.44, p = 0.8), number of arch and pumps (KW(x) = 0.37, p = 0.83, n = 52). While male size did not appear to affect the course of mating trials, mating time and the number of arch and pumps significantly increased with female size relative to other females in the experiment. On average, larger females (see Supplemental Information 4) spent significantly more time mating (KW(x) = 6.7, p = 0.03, n = 52) and had significantly more arch and pumps (KW(x) = 8.38, p = 0.01, n = 52), whereas resting (KW(x) = 3.36, p = 0.18, n = 52) and fighting (KW(x) = 1.08, p = 0.58, n = 52) were not significantly impacted by female size.

Do females exhibit mate choice?

Females were significantly more likely to initiate mating with males introduced earlier in the trials. There were 9 experimental females with at least one trial in which no mating occurred. Neither male order (χ2, p = 0.79, n = 18), male size relative to the female (χ2, p = 0.53, n = 23), male size relative to other males (χ2, p = 0.98, n = 23), nor female size relative to other females (χ2, p = 0.39, n = 23) were significant in predicting a failure to mate. However, in 13 of the 46 trials where mating behavior occurred, the female was the one to approach the male to begin mating (χ2, p = 0.003, n = 46), by either moving herself under the male or grabbing the male to pull him on top of her. This behavior was exhibited by 9 of the 24 experimental females. Significantly different from expectations, eight of these instances occurred with the first male introduced to the female, 5 with the second male, and zero with the third (χ2, p < 0.01, n = 27). The size of the male relative to the female did not appear to be a factor in whether the female would display this behavior; it occurred 6 times when the male was larger, 5 times when the male was smaller and twice when the male was approximately equal in size to the female (χ2, p = 0.34, n = 27). Neither did the size of the female appear to be a factor in this behavior; 2 small, 3 medium, and 4 large females approached the male to initiate mating (χ2, p = 0.62, n = 27). Six of the nine females that exhibited this behavior laid eggs soon after the trials were concluded.

Thirteen of the 24 experimental females, in 19 different mating trials, were observed removing an intact sperm packet. While we observed females removing sperm packets during mating trials, none of the male traits we tested showed significant correlations to this behavior. Removal happened either by the female exhaling forcefully and expelling the spermatophore (32 instances total), or the female moving her arms close to the mantle opening and “pulling” out the sperm packet (2 observations). In each instance, the removal occurred while the male was in the tank with the female. There did not appear to be any pattern among mate order or size to this observed behavior. In 8 instances, females removed sperm packets from the first male, 7 instances from the second male and 4 from the third (χ2, p = 0.61, n = 24), male regardless of size (χ2, p = 0.81, n = 24).

Genetic analyses of paternity of broods

Multiple paternity was confirmed in all experimental broods, and all but one of the non-experimental broods when analyzed manually with the conservative single-locus minimum (SLM) method. Likewise, when analyzed with GERUD, at least 2 sires were determined for both experimental and non-experimental females, indicating multiple paternity in all broods tested (see Supplemental Information 4). Despite the universal finding of multiply mated females in this experiment, fmm calculated an expected frequency of multiple mating in the population at only 37% (fmm unequal skew = 37% [2%–90%], fmm equal skew = 37% [1%–93%]).

Analysis of the parentage of broods in CERVUS showed a trend of first mating precedence in experimental egg fertilizations (Fig. 3 and Supplemental Information 4); females used significantly more sperm from the first male to mate (χ2 95% CI; p <  0.01, n = 8, χ2 80% CI; p = 0.01, n = 8). Likewise, the number of offspring sired by first males differed significantly (p < 0.01) from the number sired by last males to mate (Fig. 3), but there was no pattern of male dominance within egg strands. Multiple males were found to have sired offspring within each strand tested, and distribution of paternity among strands appeared to be random.

Figure 3 Role of mate order in determining number of offspring sired.

Percentage of eggs sired vs. male order in mating trials with Octopus oliveri, before rerunning genotypes of unassigned eggs. Here, group 0 refers to all wild males lumped into a single category as “other” mating prior to the experiment which is the highest proportional fertilization success among tested egg masses. p < 0.01 for all between 0, 2 and 3, Residual Std. Error, 0.19, df = 23, R2 = 0.51. *, significant.

When the eggs unassigned to a known male were rerun in GERUD and CERVUS, the category of “other” was split up into much smaller subsets (see Supplemental Information 4). The number of fathers that accounted for the unassigned eggs ranged from 2 to 6, suggesting that these females had mated prior to being brought into the lab for experimental trials. Rerunning the parentage analysis with these males considered shows a significant difference (p < 0.01) in the proportion of eggs sired by the wild males and first experimental males versus the second and third experimental males (Fig. 4).

Figure 4 Role of mate order in determining number of offspring sired.

Percentage of eggs sired versus male order in mating trials with Octopus oliveri after GERUD was rerun on “Other” males from putative matings that occurred in the field before collection. Here, group 0 refers to wild males separated by likely genotype into individuals, which partitions the pre-experiment mating among several individuals and reduces the mean success of each relative to the lumped “other” category presented in Fig. 2. p > 0.001 for both 0 and 1. Residual Std. Error, 0.15, df = 44, R2 = 0.28. *, significant.

The marginal likelihood ratio tests help to visualize patterns (Fig. 5) in male–female behavior and the proportion of eggs sired based on: male mating order, male body mass, number of arch and pumps during each trial, time spent mating during a trial, the number of instances where a female was seen removing a sperm packet in a trial, and the male size relative to the female (smaller, equal, or larger). Running an ANOVA and plotting each of the variables alone against the percentage of eggs sired showed a positive correlation in the size of the male in grams (p < 0.001), the number of arch and pumps in a trial (p < 0.05), and the removal of sperm packets during a trial (p < 0.05) with paternity. In contrast, mate order (p < 0.001) and mating time (p < 0.01) show significant negative correlations with the percentage of eggs sired by that male, whereas there was no relationship was detected in the relative size of males to females. Thus, the largest females might be most attractive to males, but male size has no bearing on whether mating would occur. Analyzing the data in this way might cause overfitting of the model, especially given the small sample size and large number of parameters, so we also used AIC to determine which variables were the most useful predictors. Using AIC, only the male order and body mass were included as predictive variables in the best model (Table 2 and Supplemental Information 4).

Figure 5 Explanatory variables in paternity analyses.

Single linear regression/one-factor ANOVA plots of possible explanatory variables in paternity analysis. Male mating order (likelihood ratio x2 = 23.3, df = 2, p < 0.001), Male size in grams (likelihood ratio x2 = 11.8, df = 1, p < 0.001), Number of arch and pumps observed in mating trial (likelihood ratio x2 = 3.8, df = 1, p < 0.05), Mating time in seconds (likelihood ratio x2 = 5.8, df = 1, p < 0.01), Number of times a female removed a sperm packet (likelihood ratio x2 = 5.1, df = 1, p < 0.05), Male size relative to female (l: large, m: medium, or approximately equal to female size, s: small) (likelihood ratio x2 = 3.9, df = 1, p = 0.13).

Table 2 Factors resulting in greater proportion of offspring sired by male O. oliveri.

Variable importance in the proportion of offspring sired among males of Octopus oliveri who sired multiple paternity broods; the sum of the weights of all models that include a variable (see Supplemental Information 3 for complete AIC model selection table).

	Male order	Size of male (g)	Number of arch and pumps	Mating time (sec)	Removal of sperm packet	
Importance:	0.88	0.82	0.21	0.13	0.12	
N containing models	16	15	16	15	16	

Discussion

Mating behavior of Octopus oliveri

In general, the mating behavior of Octopus oliveri appears typical of other published reports in the genus (Mangold, 1987; Forsythe & Hanlon, 1988). The only deviation of note is the beak-to-beak mating, which although observed was still relatively uncommon (∼25%). Rodaniche (1984) was the first to describe beak-to-beak mating in the larger Pacific striped octopus; in his observations, however, beak-to-beak was the only mating position exhibited by that species. In Octopus oliveri, the mount, reach, and beak-to-beak mating positions were all observed for the first time in a single species, and parentage confirms that all positions can result in successful fertilization for this species.

Beak-to-beak mating is considered a dangerous position for the male, because sexual cannibalism has been observed in a number of octopus species (Hanlon & Forsythe, 2008). Cannibalism did occur among non-experimental Octopus oliveri when housed in a large communal tank but it was unclear if it was sexual cannibalism, competitive, or for other reasons. No cannibalism was observed in any of the experimental mating trials but that does not rule out the possibility that it might occur in the wild, and the fact that cannibalism occurs in communal tanks suggests that males might be wary of mating in a position that would make them vulnerable to consumption. This risk might account for the relative rarity of beak-to-beak mating. Still, our results indicate such mating happens successfully, because of the nine females who had trials where beak-to-beak mating occurred, five subsequently laid eggs. Chemical cues have been found to be important in squid, octopus, and cuttlefish mating (Buresch et al., 2004; Boal, 2006; Cummins et al., 2011; Polese, Bertapelle & Di Cosmo, 2015; Morse et al., 2016), so it is possible that the males in these trials might have been responding to a chemical cue from the female inviting more risky mating behavior.

Larger females tended to incite longer mating durations with higher numbers of arch and pumps by males. Size in octopuses is generally dependent on environmental factors such as food quality and temperature and it can therefore be difficult to determine what size determines sexual maturity in a female (Semmens et al., 2004). However, in some octopuses size can be a predictor of maturity and fecundity, which might indicate that males are more likely to invest time in mating with larger females (Huffard, Caldwell & Beneka, 2008; Leporati, Pecl & Semmens, 2008; Mohanty, Ojanguren & Fuiman, 2014). In the case of Octopus oliveri, it also appears that larger females are more amenable to mating, possibly because they are closer to spawning, or perhaps because they produce more eggs to fertilize. While it is well known that female octopuses can mate and store sperm months before laying eggs (Wells, 1978; Anderson, Mather & Wood, 2013), it might be that the quality of the sperm is reduced over time (Reinhardt, 2007), making it likely that smaller females would be more likely to delay mating until they are closer to spawning.

Female choice in mating

Initially, we interpreted the first approach by females and sperm removal as evidence of female choice. However, both behaviors might be better explained by alternative hypotheses. While it was a relatively rare occurrence for females to approach males for mating (∼28%), it is significant that in more than 60% of these cases, it was the first male presented to the female, regardless of size difference (Fig. 3). This preference might indicate that mature females isolated from males would be more responsive to mating with the first male that is presented to them. If so, that would suggest that male encounter rate in the wild is not so high as to avoid sperm limitation, and that multiple mating might be a strategy to avoid reduced fertilization rate. Therefore, the first approach by females might not be choice, but rather desperation due to sperm limitation. Clearly studies in the field to observe this octopus mating would be beneficial, but field observations of this species are rare and extremely difficult because they live exclusively in high wave action areas with dangerous rocky terrain and are nocturnal.

A previous study of Octopus bimaculoides mating behavior found a similar pattern to that in this study, with large females mating for longer periods with the first male to approach them (Mohanty, Ojanguren & Fuiman, 2014). But, as with our results, it might be possible that as more mates are presented to them, females might become more selective. Also, of the nine females who exhibited primary approach behavior, six of them laid eggs at the end of the experiments. The other three died, two during a water failure, and the last died unexpectedly, without laying eggs, but it is notable that every individual that survived successfully laid eggs. As suggested by Mohanty, Ojanguren & Fuiman (2014), if these females were nearing brooding, they might have been trying to acquire as much sperm as possible.

Active sperm removal might be a function of female choice, particularly if it were also a signal to the male that the female was not receptive to mating. However, it might also be simply mechanistic, which is more likely the case in this study. Wodinsky (2008) reports on mating of two Octopus species and noted that females were seen to expel spermatophores before the spermatozoa within the spermatophore had ejaculated. He concluded it was a result of a disconnection between the calamus (the tip of the hectocotylus) and the distal oviduct. If this disconnection were the cause, the observed active removal of sperm packets would have nothing to do with female choice. Given that there was no pattern among male size or precedence in incidences where the females were observed to remove sperm packets, it would suggest that any male could place the ligula incorrectly. In this case, if the sperm removal were mechanistic, it would indicate that no female choice is occurring; rather it is a function of clearing the passage to the oviduct to allow further spermatophores to be transferred. Given that we found a positive correlation between the occurrence of sperm removal and paternity it suggests that this behavior is indeed a result of misplaced spermatophores and not a function of choice, because a negative correlation between sperm removal and paternity would be expected if this behavior were a function of choice.

Multiple paternity

Our data confirm multiple paternity in Octopus oliveri in every experimental mating that we conducted, but fmm suggests that the rate is only moderate in the field (fmm unequal skew = 37% [2%–90%], fmm equal skew = 37% [1%–93%]). Added to reports of multiple paternity in the deep sea octopus Graneledone boreopacifica (Voight & Feldheim, 2009), the shallow water Octopus vulgaris (Quinteiro et al., 2011), and the long-armed octopus Octopus minor (Bo et al., 2016), it appears that this reproductive outcome is the norm among octopods. The fact that females tend to initiate mating most often with the first male to which they are introduced, and then become more choosy as more mates are provided suggests that sperm limitation might be a reasonable explanation. Likewise, if larger females were more fecund, that would be consistent with the tendency for larger females to encourage prolonged mating time and increased numbers of arch and pumps (and therefore more spermatophores) during mating.

Sperm competition and mating precedence

It has been reported that the ligula on the tip of the hectocotylus may be used to remove sperm deposited by previous males (Cigliano, 1995; Quinteiro et al., 2011). Unlike Cigliano’s experiments in 1995, our experiments did not show any evidence of a sperm competition mechanism between males. There was no significant change in the amount of time between when the male would insert his hectocotylus and when he would begin the arch and pump movements, regardless of the mate order, absolute or relative body size. In addition, and in contrast to what was found in the squid Loligo vulgaris reynaudii (Shaw & Sauer, 2004), there was no clear distribution of sires among the individual strings or among the whole brood in O. oliveri.

Such differences might result from variation among species in the tissue of the ligula and calamus of octopuses that could impact the ability to displace spermatozoa of previous males (Voight, 2002; Thompson & Voight, 2003). For example, the ligula of O. oliveri is very short and lacks flexibility (Garcia, 2010), so perhaps this limits their ability to remove sperm deposited by previous males. Alternatively, the time between mating sessions might have been sufficient to allow spermatozoa to penetrate deep into the spermatheca (De Lisa et al., 2013), therefore rendering sperm removal difficult or impossible. For example, spermatozoa have been found in the oviducal gland of O. tetricus one day (24 h) after mating, although whether it was the sperm of the experimental male or a previous male from the field was unclear (Joll, 1976).

Studies examining precedence in cephalopods have focused predominantly on loliginid squids and cuttlefish (Buresch et al., 2009; Voight & Feldheim, 2009; Quinteiro et al., 2011; Sato, Yoshida & Kasugai, 2016). Last male precedence was found in two squid species: Loligo bleekeri and Loligo vulgaris reynaudii (Shaw & Sauer, 2004; Iwata & Munehara, 2005), and one cuttlefish species: Euprymna tasmanica (Squires et al., 2015). In contrast, no clear precedence was found in the cuttlefish Sepia apama (Naud et al., 2004). In both squid and cuttlefish, males can deposit sperm packets (spermatangia) either inside the mantle, or around the buccal mass surrounding the mouth, which leads to both external and internal fertilization. Possibly because many (but not all) species of squid and cuttlefish mate in large aggregations, the last male to encounter the female can ensure paternity by guarding the female while eggs are laid. In contrast, octopuses have only internal fertilization, and contrary to last male precedence commonly reported in squids and cuttlefish, we found early male precedence among our experimental mating trials in Octopus oliveri. However, this is not first male precedence because none of the females collected were likely to be virgins, and the relative contribution of matings prior to the start of the experiment cannot be accurately determined. Nonetheless, there is skew among every brood tested, indicating that some males are fertilizing more offspring than others, and among our experimental mating trials, the last males clearly sired significantly fewer offspring than the first males (Fig. 3). One possible explanation for why the first males in our experiments did so well in terms of fertilization success is that females captured for this experiment might have been storing sperm for long enough that it had decreased in quality. When presented with a new male at the outset of the experimental matings, these males might have been able to displace or overwhelm the low-quality sperm of previous males. If this were the case, it is also possible that the first male could have overwhelmed the spermathecae, making sperm depositions by subsequent males less successful.

In addition to mating order, size was the only other predictive variable for parentage in this study. Although size did not influence the ability of a male to mate in the behavioral experiments, the use of the microsatellite markers indicates larger males sire significantly more offspring. This could be due to some factor such as an unknown mechanism of differential use of deposited sperm by the female, but we suspect that large males have larger spermatophores and are sperm loading, or overwhelming the spermathecae with their sperm (Simmons & Fitzpatrick, 2012). There was no significant correlation between the number of arch and pumps and the number of offspring sired, but larger males might contribute larger spermatophores containing higher numbers of individual spermatozoa, thereby increasing their chances of fertilization over smaller males. Among octopodids studied to date, spermatophore size tends to be highly correlated with mantle length (Mann, 1984). Each spermatophore contains a sperm reservoir, which contains the individual spermatozoa. Voight (2002) found that sperm reservoir length is very tightly correlated with spermatophore length, suggesting that males are incapable of manipulating the size (and therefore the number of spermatozoa) of the spermatophore to maximize the amount of sperm delivered to the female. Although spermatophore size was not measured in this study, we see only a correlation between male body mass, not numbers of arch and pumps, in successful paternity of broods, leading us to hypothesize that larger males might transfer more spermatozoa than smaller males. Further research is needed to determine if sperm loading might be a sperm competition strategy in octopods.

Conclusion

These experiments indicate that females of Octopus oliveri appear to mate indiscriminately with males in any order and of any size, showing minimal behavioral evidence for pre-copulatory sexual selection. Successful mating occurred in each of the mount, reach and beak-to-beak positions, and larger females elicited greater mating effort from males in terms of duration and arch and pump behaviors. Multiple paternity was observed in every experimental cross when females were presented with 3 potential mates under laboratory conditions but was estimated to be comparatively low in the field. This low population rate of multiple paternity might indicate sperm limitation due to rare mate encounter in the field and could explain both female responses to first males in our behavioral assays and early male advantage in parentage of broods. We see no evidence of direct sperm competition in Octopus oliveri, but larger males produce significantly more offspring, perhaps because they can include more spermatozoa in spermatophores. This study contributes to the growing research on cephalopod mating systems and indicates that octopus mating dynamics might be more variable and complex than thought previously.

Supplemental Information

Data S1 Raw data from manuscript for Octopus oliveri

Raw data for all behavioral observations, weights and measurements performed in this study.

Click here for additional data file.

Supplemental Information 2 R code for the analyses of mating behavior in Octopus oliveri

R code for each of the statistical analyses performed in this study.

Click here for additional data file.

Supplemental Information 3 FASTA sequences for the 48 microsatellite loci tested for this study

Raw sequence data from which microsatellite loci described in this study were designed. We present all 48 loci discovered from our sequencing efforts and include primers tested in the supplementary materials, although only a subset were used in this study.

Click here for additional data file.

Supplemental Information 4 Supplementary figures from the manuscript and microsatellite primer sequences

Additional figures, analyses and a complete list of microsatellite loci and primers discovered for Octopus oliveri to support the presentation of data in the primary manuscript.

Click here for additional data file.

We thank the ToBo lab, especially Zac Forsman and Ingrid Knapp, for their assistance and advice with the genetics component of this research, and to Jeff Drazen, Les Watling & Chuck Birkeland for their support and feedback throughout this project. This is contribution #1757 from the Hawai‘i Institute of Marine Biology, and SOEST #10709.

Additional Information and Declarations

Competing Interests

Author Contributions

Animal Ethics

Field Study Permissions

DNA Deposition

Data Availability

Robert J. Toonen is an Academic Editor and the Section Editor for Aquatic Biology at PeerJ.

Heather Ylitalo conceived and designed the experiments, performed the experiments, analyzed the data, prepared figures and/or tables, authored or reviewed drafts of the paper, approved the final draft.

Thomas A. Oliver conceived and designed the experiments, analyzed the data, prepared figures and/or tables, authored or reviewed drafts of the paper, approved the final draft.

Iria Fernandez-Silva and Robert J. Toonen conceived and designed the experiments, analyzed the data, contributed reagents/materials/analysis tools, prepared figures and/or tables, authored or reviewed drafts of the paper, approved the final draft.

James B. Wood conceived and designed the experiments, analyzed the data, authored or reviewed drafts of the paper, approved the final draft.

The following information was supplied relating to ethical approvals (i.e., approving body and any reference numbers):

This work was completed in 2012, prior to when IACUC approval was needed for cephalopod animals at our institution.

The following information was supplied relating to field study approvals (i.e., approving body and any reference numbers):

Hawai‘i State Department of Land and Natural Resources, Division of Aquatic Resources

The following information was supplied regarding the deposition of DNA sequences:

Data is available at GenBank, accession numbers: KC848885 and KC848886.

Additional sequence data from which microsatellite loci were developed is available as a Supplemental File.

The following information was supplied regarding data availability:

The raw data is available as Supplemental File.

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
