# Peer review of "A behavioral and genetic study of multiple paternity in a polygamous marine invertebrate, Octopus oliveri"

_PeerJ, doi:10.7717/peerj.6927_

## Round 0.1 · original submission · Major Revisions

The reviewers found the paper to be an interesting and potentially valuable contribution to the literature. Reviewer 2 provides several useful comments for clarifying the methods. Two of the reviewers find some grammatical problems; I recommend a careful reading to catch mistakes and writing problems before you resubmit. Note that Reviewer 3 has attached their review as a PDF, so please be sure to read these comments.

·

Basic reporting

no comment in general but I suggest to add these references and add in the introduction some issues about the stimuli that can guide the animals in the choice of the partner Frontiers in Physiology
Volume 7, Issue SEP, 28 September 2016, Article number 434
Neuroendocrine-immune systems response to environmental stressors in the cephalopod Octopus vulgaris(Short Survey)
Di Cosmo, A.Email Author, Polese, G.

Role of olfaction in Octopus vulgaris reproduction Polese, G., Bertapelle, C., Di Cosmo, A. 2015 General and Comparative Endocrinology
210, pp. 55-62

Olfactory organ of Octopus vulgaris: Morphology, plasticity, turnover and sensory characterization

Polese, G., Bertapelle, C., Di Cosmo, A. 2016 Biology Open
5(5), pp. 611-619

Experimental design

no comment

Validity of the findings

no comment

Additional comments

My comment and suggestions to the AA is fo add major informations and references about what are the stimuli that can guide the animals in the choice of the partner.

Reviewer 2 ·

Basic reporting

"Who's your daddy" absolutely must be removed from the title. The use of this phrase is unprofessional and tone deaf at best, as it can be HIGHLY offensive to female readers. Look up the phrase in Urban Dictionary or Wikipedia to learn what it means to most people in the US.

Experimental design

While the experimental design appears to be sufficient, sample sizes need to be explained more clearly both in the methods and the results. The authors do not address how the reuse of males in different trials was treated in the statistical analyses. By using the same males with different females, the replicates were not independent, and it's possible assumptions of the statistical tests could have been violated. Please clarify.

Animal sizes (wet weight of individuals, and relative masses of mating pairs) were recorded but not reported given, which is unconventional for mating studies conducted in the lab.

Data are not made available, though this is not unusual for mating experiments.

Validity of the findings

no comment

Additional comments

Overall this is an interesting study about the mating behavior and fertilization patterns in Octopus oliveri, and an important paper in the field of cephalopod behavior. It contributes to our understanding of the complex relationships between mating behavior and fertilization patterns, which is a very new avenue of research for cephalopods. Overall this study appears to be well designed, though important details will need to be included in the revision, because it's possible the experimental design limits the statistical analyses that can be used appropriately.

In the detailed comments below I have suggested papers the authors might like to read to provide additional context for interpreting their results, in particular the similar studies by Morse. The authors might also find Itawa's works on squids relevant to their discussion of sperm competition.

Detailed comments:

Abstract:
While it becomes more clear later in the text, as it's worded here, it's not clear whether the entire study is based on the same three males and six females, or whether each experiment had its own 3 males and 6 females. Please clarify. It's not clear from the abstract (or the main text) what the four sets of experiments were. Were they separate experiments or were they replicates? If they were replicates, did they use the same or different individuals?

Line 27: Replace find with found. Here and throughout, all methods and results should be given in past tense, not present or imperfect.

Line 30: Replace may with might here and throughout.
Are the female sperm limited if the eggs are still fertilized? Maybe the wild females are guarded.

Line 32/33: I'm curious to see how you showed this in this experiment. Otherwise it is speculation that should not be included here.

Introduction paragraph 2: Sneaker males were described for Octopus cyanea in 1997. Other relevant references: Cigliano, J. A. (1995). Assessment of the mating history of female pygmy octopuses and a possible sperm competition mechanism. Animal Behaviour, 49(3), 849-851.
Morse, P., Zenger, K. R., McCormick, M. I., Meekan, M. G., & Huffard, C. L. (2015). Nocturnal mating behaviour and dynamic male investment of copulation time in the southern blue-ringed octopus, Hapalochlaena maculosa (Cephalopoda: Octopodidae). Behaviour, 152(14), 1883-1910. Morse, Peter, Christine L. Huffard, Mark G. Meekan, Mark I. Mccormick, and Kyall R. Zenger. "Mating behaviour and postcopulatory fertilization patterns in the southern blue-ringed octopus, Hapalochlaena maculosa." Animal Behaviour 136 (2018): 41-51.

Line 54: Replace males with male's and another with another's sperm

Line 62: Cigliano's work showed behavioral evidence of sperm competition, but no evidence of sperm precedence.

Discuss this work here, but word accordingly.

Line 68: Another relevant work to consider reading is Morse, Peter, Christine L. Huffard, Mark G. Meekan, Mark I. Mccormick, and Kyall R. Zenger. "Mating behaviour and postcopulatory fertilization patterns in the southern blue-ringed octopus, Hapalochlaena maculosa." Animal Behaviour 136 (2018): 41-51.

Lines 72-75: Redundant with previous paragraph

Lines 91-108: This information should be incorporated into the introduction.

Lines 111, 116: Give lat and long of collection site, and full location (state, country) for both.

Line 115: Sample sizes are not described clearly. Give total number of animals collecton. Are the same 6 + 3 used for all trials, or different 6 + 3 sets used in each of the four experiments? I assume different, but use/reuse of individuals is not entirely clear.

Line 119: Delete HIMB because this acronym is not used elsewhere.

Line 122: Spell out NOAA

Somewhere in the methods or acknowledgements, mention the collection/research permit situation (agency, issued to whom)

Lines 130-135: add a diagram showing sample sizes/uses of individuals in experiments

Line 132: Describe the three experiments. The abstract mentions four experiments.

Line 134: Give actual sizes of males and females.

Lines 150-154: Do you know of any work looking into spermatophore regeneration, because that's what is most important here? Perhaps from squid work?

Line 155: replace with "The trial history of each female was recorded to..."

Line 159: It's not clear how the experimental design yielded sixty-two trials. 3 males x 4 females x 4 experiments (abstract) or 3 experiments (methods). Similarly, it's not clear what the final sample sizes were for the statistical tests.

Line 179: Would it be more accurate to say that fighting was followed by mating? Or were these examples of forced copulation in your view? Please specify.

Line 192: Clarify in the diagram requested above how these DNA-sampled individuals fit in with the experiments described above.

Line 196: Do you mean "and" instead of "or"?

Results section: Consider moving the behavioral data analysis section to before the paternity analysis section.

Lines 231-232: This is a lot of factors for such a small sample size.

Lines 237-238: These sample sizes are helpful. Please list them for all tests.

Line 242: List the R version and cite the software.

Lines 291-293: Interpretations should be moved to the discussion.
Replace the citation of Huffard 2007 with the more appropriate Huffard, C. L., Caldwell, R. L., & Boneka, F. (2008). Mating behavior of Abdopus aculeatus (d’Orbigny 1834)(Cephalopoda: Octopodidae) in the wild. Marine Biology, 154(2), 353-362.

Line 300: Interesting!

Lines 300-302: This is a bit redundant with methods.

Line 314: The data...were...(datum is the singular).

Figures: Include either Figure 1 or Figure 2 (and accompanying results) based on which you think gives the most accurate representation of your experimental results.

Lines 415-422: Will you publish a general account of these behaviors and how common they were? Did beak-to-beak mating also involve enveloping of the male by the female as described in Caldwell, R. L., Ross, R., Rodaniche, A., & Huffard, C. L. (2015). Behavior and body patterns of the larger pacific striped octopus. PloS one, 10(8), e0134152? What is it about these two species that leads to beak-to-beak mating when it's so rare otherwise?

Lines 433-443: Strong evidence of males investing more mating effort in larger females, and time spent by females mating, is also discussed here: Huffard, C. L., Caldwell, R. L., & Boneka, F. (2008). Mating behavior of Abdopus aculeatus (d’Orbigny 1834)(Cephalopoda: Octopodidae) in the wild. Marine Biology, 154(2), 353-362.

Lines 453: Earlier in the paper you mention analyzing the broods of females caught but not mated in experiments. These serve as your controls for this discussion on sperm limitation. Please present the results of those paternity runs more thoroughly. I lost track of them. They are an important part of your study and should be highlighted.

Lines 446-466: Consider combining and streamlining the first two paragraphs of the female mate choice section.

Lines 518 and 523: Replace "In contrast" with "By contrast"

Line 531: Replace "our first males" with "the first males in our experiments"

Lines 437-438: This statement is not consistent with the results presented in lines 401-401

Reviewer 3 ·

Basic reporting

no comment

Experimental design

no comment

Validity of the findings

no comment

Additional comments

In the post-copulatory sexual selection (sperm competition and cryptic female choice), it is important to know how many males copulate with a female and what is important to decide the fertilization success. Sperm precedence is one of the important factors in the fertilization and last copulated males sired more offspring in general “refer to last male precedence”.
Cephalopods have various sperm storage organs and mechanisms in their reproduction, which suggest the reproductive traits originally evolved through post-copulatory sexual selection. Nevertheless, there are few studies to exam what are important to decide their reproductive success, especially in Octopus.
This study conducted a sequential mating experience in Octopus oliveri and showed mating order did not relate with fertilization success. This result would contribute to the knowledge of cephalopod reproduction. And one behavioural report, spermatangia reject by females, is important for considering cryptic female choice. However, I have to write the quality of this manuscript is quite low. The authors should check their text at least twice before submitting because their manuscript has so many grammatical and logical problems. They also have to study more in cephalopod reproduction, sexual selection and statistics. There were many problematic descriptions throughout. At first, they have to read a recent paper written by Morse et al (2018) which is a study of paternity analysis in blue-ringed octopus. This paper would be a good sample to make their draft. The authors should re-submit it considering what is a new for their study. Unfortunately, I cannot recommend to accept this manuscript without major revision.

Annotated reviews are not available for download in order to protect the identity of reviewers who chose to remain anonymous.

---

## Round 0.2 · accepted · Accept

Your paper will make an interesting and useful contribution to the literature. The reviewer makes a good point about the use of the word "daddy" in the title, but I will leave the final decision about the title to you.

# Reviewer 2 ·

Basic reporting

no comment

Experimental design

no comment

Validity of the findings

no comment

Additional comments

Thank you for tracking changes in the review file. This is very helpful. These changes improve the manuscript. Great paper.

You should remove the “Who’s the daddy” question from the title for a few main reasons. This question makes no intellectual contribution to the title. As catchy as the phrase might sound in some circles, many people in the world will have no understanding of the tongue and cheek reference you make, and the use of the colloquial term daddy risks confusing or alienating people who do not use English as their first language. It’s best to stick to the terms most often used in the scientific literature for this reason. On the other hand, some people will get this reference, and you risk offending these readers. It’s best to omit, as you are open to doing.

Line 179: Provide a citation for this literature.